# Reactive Oxygen Species Signaling and Oxidative Stress: Transcriptional Regulation and Evolution

**DOI:** 10.3390/antiox13030312

**Published:** 2024-03-01

**Authors:** Yuhang Hong, Alessandra Boiti, Daniela Vallone, Nicholas S. Foulkes

**Affiliations:** Institute of Biological and Chemical Systems, Karlsruhe Institute of Technology, 76344 Eggenstein-Leopoldshafen, Germany; yuhang.hong@kit.edu (Y.H.); alessandra.boiti@kit.edu (A.B.); daniela.vallone@kit.edu (D.V.)

**Keywords:** reactive oxygen species, cellular signaling, DNA repair, transcriptional regulation, vertebrate evolution

## Abstract

Since the evolution of the aerobic metabolism, reactive oxygen species (ROS) have represented significant challenges to diverse life forms. In recent decades, increasing knowledge has revealed a dual role for ROS in cell physiology, showing they serve as a major source of cellular damage while also functioning as important signaling molecules in various biological processes. Our understanding of ROS homeostasis and ROS-mediated cellular signaling pathways has presumed that they are ancient and highly conserved mechanisms shared by most organisms. However, emerging evidence highlights the complexity and plasticity of ROS signaling, particularly in animals that have evolved in extreme environments. In this review, we focus on ROS generation, antioxidative systems and the main signaling pathways that are influenced by ROS. In addition, we discuss ROS’s responsive transcription regulation and how it may have been shaped over the course of evolution.

## 1. Introduction

The evolution of aerobic respiration represented a significant milestone in the history of life on Earth. It is a metabolic process that enables organisms to efficiently extract energy from complex organic compounds by the use of an electron transport chain in the presence of oxygen as a final electron acceptor [1]. As organisms evolved to exploit the benefits of aerobic respiration, a major challenge emerged due to the accumulation of reactive oxygen species (ROS), inevitable by-products of cellular aerobic respiration, including the superoxide anion (O_2_•^−^), hydrogen peroxide (H_2_O_2_) and the hydroxyl radical (HO•) [2]. ROS serve both as potent sources of macromolecular damage as well as essential signaling molecules and so maintaining a balance between these negative and positive effects is fundamentally important for normal cellular physiology. These basic properties of ROS have been extensively documented in many previous reviews [2,3,4]. However, in this current review we aim to broaden the view of the role of ROS in cell biology and physiology by also exploring species-specific differences in the mechanisms which react to and regulate ROS as well as speculating how the environment may have shaped these mechanisms over the course of evolution.

## 2. The Origins of ROS and Oxidative Stress

As the powerhouse of the cell, mitochondria consume approximately 90% of the body’s oxygen to generate ATP through oxidative phosphorylation, rendering them a significant source of ROS [5]. On the inner membrane of mitochondria, the superoxide anion (O_2_•^−^) is primarily produced in complexes I or III of the electron transport chain as a result of the monoelectronic reduction of O_2_. Then a series of enzymatic reactions comprise the metabolic pathways for the processing of endogenously generated ROS during aerobic respiration [6]. ROS species represent chemically reactive molecules which are highly toxic since they induce damage in various cellular macromolecules including lipids, proteins and nucleic acids [7], demonstrating that there is a high price to pay for the use of O_2_ to enhance energy production. As a consequence, ROS participate in various pathological processes in almost all aerobes [8].

Oxidative stress is a physiological condition characterized by an imbalance between the production of ROS and the ability of cells to detoxify them [9]. It plays a significant role in the disruption of normal cellular function and can lead to inflammation, tissue injury and organ dysfunction. As shown in Figure 1, while cellular ROS can be generated endogenously in the mitochondria as the primary intracellular site for oxidative phosphorylation, other organelles linked with ROS production include the peroxisome where several peroxisomal enzymes serve to generate and metabolize ROS [10], as well as the endoplasmic reticulum where misfolded protein aggregation leads to “ER stress” and an associated increase in ROS [11]. Production of ROS by metabolic enzymes such as the NADPH oxidases (NOX) represents another important source of oxidative stress [7,12]. Furthermore, excess ROS may be induced by exposure to sunlight or xenobiotic compounds [3]. Accumulation of ROS leads to cellular damage by either direct reaction with biological molecules or the indirect regulation of signaling pathways [13,14]. Thereby, oxidative stress has been implicated in the progression of cancer by promoting DNA mutations and altering cell signaling pathways [15]. Furthermore, it has been implicated in the aging process, as accumulation of oxidative damage over time contributes to the decline in cellular function and the development of age-related diseases. Therefore, understanding and managing oxidative stress represent important challenges for biomedicine.

## 3. Intracellular ROS Balance and Antioxidant Systems

Due to the high toxicity and carcinogenic effect of ROS, the maintenance of ROS levels within well-defined limits is a critical facet of cellular physiology, with profound implications for health and disease. Several ROS-based redox regulatory pathways govern intracellular ROS homeostasis, with the ROS-mediated oxidation of cysteine residues being among the most extensively studied mechanisms [2]. Cysteine residues serve as specific targets for ROS and constitute the catalytic centers of many ROS scavenger proteins, such as thiol peroxidases. For example, peroxiredoxins (Prxs) contain conserved cysteine residues within their catalytic sites, which are susceptible to oxidation by H_2_O_2_. Upon exposure to H_2_O_2_, the peroxidatic cysteine residue of Prxs undergoes oxidation to form a sulfenic acid intermediate. Subsequent reversible reactions with “resolving” cysteine residues within the same or neighboring Prx subunits result in the formation of disulfide bonds or higher oxidation states. These oxidative modifications can alter the activity and oligomeric state of Prxs, thereby modulating their peroxidase activity and interaction with downstream signaling molecules [3]. In addition, another crucial oxidative modification involves methionine residues. When exposed to ROS, the majority of methionine residues undergo conversion to methionine sulfoxides, resulting in structural and functional modifications of proteins [16]. With the help of methionine sulfoxide reductase, methionine oxidation by ROS becomes a reversible process and this protects the integrity and stability of proteins from oxidative damage.

### 3.1. Cellular Antioxidative Systems

Cells employ a multifaceted antioxidant defense system to regulate ROS levels and prevent oxidative stress. This system includes a network of enzymatic and non-enzymatic antioxidants that work in concert to neutralize ROS (Figure 2) [17]. Enzymatic antioxidants are mainly represented by enzymes such as superoxide dismutase (SOD) and catalase (CAT). SOD constitutes the primary line of antioxidative defense, playing a pivotal role by catalyzing the dismutation of superoxide anions (O_2_•^−^) to form H_2_O_2_ [18]. The subsequent detoxification of H_2_O_2_ into water and oxygen is catalyzed by enzymes including CAT, glutathione peroxidase (GPx) and peroxiredoxins (Prxs) [19]. In addition, the glutathione-associated enzymes such as glutathione reductase (GR) and glutathione S-transferase (GST) participate in glutathione metabolism and constitute a secondary level of defense [20]. Upon exposure to ROS, the majority of methionine residues undergo conversion to methionine sulfoxides, resulting in structural and functional modifications of proteins [16]. With the help of methionine sulfoxide reductase, methionine oxidation by ROS becomes a reversible process and this protects the integrity and stability of proteins from oxidative damage. Other proteolytic enzymes and DNA repair enzymes which remove lesions from protein or DNA have also been identified as important antioxidants, which indirectly function in cellular redox balance [21,22].

Non-enzymatic antioxidants encompass a diverse range of molecules, such as vitamins, glutathione and metal-binding proteins like ferritin, which quench ROS directly or participate in the regeneration of enzymatic antioxidants [23]. Ascorbic acid, for example, commonly known as vitamin C, serves to neutralize free radicals. The antioxidative properties of vitamin C are primarily attributed to its ability to readily donate electrons, stabilizing and thereby quenching the damaging oxidative reactions initiated by ROS [24]. Furthermore, vitamin C has been shown to regenerate other important antioxidants, such as vitamin E, thereby enhancing the overall antioxidant defense network [25]. This micronutrient’s versatile role extends beyond direct radical scavenging, as it also modulates the activity of transcription factors involved in the expression of antioxidant enzymes, further bolstering the body’s innate defense against oxidative stress [26,27]. Moreover, vitamin C’s hydrophilic properties allow it to exert its antioxidative influence in both aqueous and lipid environments, making it a crucial player in protecting various cellular components from oxidative damage [28]. However, ascorbic acid also exhibits pro-oxidant effects under certain conditions, particularly in the presence of transition metal ions such as iron and copper. In these circumstances, ascorbic acid can undergo redox cycling, donating electrons to transition metals, and lead to ROS generation [29]. This pro-oxidant activity of ascorbic acid has been implicated in mediating the cytotoxic effects of oxidative stress and DNA damage at pharmacologic concentrations. Thereby, this property may also be beneficial for anti-tumor therapies due to the associated induction of DNA repair enzymes and selective cytotoxicity in tumor cells [30,31].

### 3.2. Reactive Sulfur Species as Antioxidants

Reactive sulfur species (RSS) represent a diverse group of sulfur-containing molecules that play significant roles in oxidative stress pathways. While traditionally overshadowed by ROS, RSS have gained increasing recognition for their involvement in redox signaling and cellular homeostasis [32]. Hydrogen sulfide (H_2_S), a well-known RSS, has emerged as a key player in modulating oxidative stress responses due to its potent antioxidant properties and regulatory effects on cellular signaling pathways. H_2_S can scavenge ROS directly or indirectly through the upregulation of antioxidant enzymes, such as SOD and CAT, thereby mitigating oxidative damage. Moreover, H_2_S can reversibly modify cysteine residues in proteins via sulfhydration, regulating protein function and redox signaling cascades [33]. Additionally, other RSS, such as hydropersulfides and polysulfides, contribute to cellular antioxidant defenses and redox homeostasis as endogenous antioxidants through similar mechanisms [34]. Although the chemical nature of RSS in various biological activities remains poorly understood, recent studies have indicated their regulatory roles in pathophysiological conditions, including cardiovascular diseases [35], neurodegenerative disorders and cancer [36].

### 3.3. Role of Nitric Oxide in Oxidative Stress

Nitric oxide (NO) is a versatile signaling molecule in biological systems, exerting both pro-oxidant and antioxidant effects depending on its concentration, cellular context and interaction with other molecules. At low concentrations, NO acts as an antioxidant by scavenging free radicals and inhibiting lipid peroxidation, thereby protecting cells from oxidative damage [37]. For example, NO can directly interact with lipid peroxyl radicals (LOO•), inhibiting the propagation of lipid peroxidation chain reactions and protecting cell membranes [38]. However, due to a lack of enzymatic scavengers, NO overproduction by inducible NO synthase (iNOS) under stress conditions can give rise to reactive nitrogen oxide species (RNOS), such as peroxynitrite (ONOO−), and so contribute to a reversal from protective to deleterious effects of NO [39]. For example, excessive NO production by iNOS in inflammatory cells has been implicated in pro-inflammatory effects during the pathogenesis of various autoimmune and chronic inflammatory diseases [29]. Therefore, the delicate balance between NO and ROS exemplifies the dynamic interplay within oxidative stress pathways, highlighting the need for a more detailed understanding of their roles in cellular physiology and pathology in order to develop effective strategies for managing oxidative stress-related diseases.

## 4. ROS Serve as Signaling Molecules

Although traditionally recognized for their potentially damaging effects, recent research has unveiled the dual role of ROS as vital signaling molecules engaged in regulating numerous biological processes [4]. At physiological levels, ROS serve as crucial signaling molecules in a multitude of cellular functions, including but not limited to cell growth, proliferation, differentiation, apoptosis, immune response and stress adaptation [40]. Once they exceed the normal physiological range, they lead to cellular damage resulting in pathogenesis (Figure 3). H_2_O_2_ is considered the best candidate to serve as a signaling molecule due to its higher stability, selective reactivity and diffusibility compared to other ROS molecules [2]. The physiological range of intracellular H_2_O_2_ concentrations seems to be conserved in various life forms, and it becomes toxic at concentrations above 0.5 × 10^−4^ M, at which it induces cell apoptosis [41]. ROS function as secondary messengers, via the modulation of the activity of numerous enzymes, transcription factors and signaling cascades [4]. Therefore, an understanding of the dual nature of ROS, as both harmful and regulatory molecules, provides essential insight into the complex interplay between cellular metabolism, oxidative stress and the regulation of biological processes. 

A combination of regulatory mechanisms ensures that ROS remain at “safe” levels but also enables their signaling functionality. Characteristic features of these ROS-responsive signaling pathways are that the function of certain “sensor” components is influenced directly by the redox state, and also that transcription factors serve as “effectors” and thereby can coordinate appropriate programs of gene expression. Over the past three decades, several pivotal ROS-responsive signaling pathways and transcription factors have been identified, showcasing their reliance on regulation by ROS [4,7,42]. In the next sections, we outline these ROS-responsive signaling pathways and illustrate how they serve as bridges between ROS and changes in gene expression.

### 4.1. AP-1 

The Activator Protein-1 (AP-1) family represents a diverse subgroup of the basic leucine zipper (bZIP) transcription factors which are crucial for the regulation of cellular responses to a myriad of extracellular stimuli including ROS. bZIP transcription factors are proteins which comprise the second largest dimerizing network found in all eukaryotes [43]. They possess a C-terminal leucine zipper domain, which enables them to form homo and heterodimers which can bind to DNA in a sequence specific manner. DNA binding is mediated by a basic amino acid-rich domain which lies adjacent to the leucine zipper. The AP-1 transcription factors are comprised of homo- and heterodimers formed by members of the Jun (c-Jun, JunB, JunD), Fos (c-Fos, FosB, Fra-1, Fra-2), Maf (c-Maf, MafA, MafB, MafG/F/K, Nrl) and ATF (ATF2, ATF3, B-ATF, JDP1, JDP2) protein subfamilies. AP-1 proteins intricately regulate gene expression by binding to specific DNA sequences, commonly known as AP-1 sites (with a core consensus sequence of “TGACTCA”) within target gene promoters [44]. The composition of AP-1 dimers imparts functional diversity, with distinct family members exhibiting unique affinities for various DNA sequences and interacting partners. These proteins play indispensable roles in fundamental cellular processes, including proliferation, differentiation and apoptosis [45]. AP-1 factors are activated by a variety of growth factors, cytokines, neurotransmitters, hormones and environmental stressors like toxins and ultraviolet light (UV). In particular, numerous studies have revealed a highly conserved property of AP-1 from yeast to mammals that it is redox-regulated, and that the activation of AP-1 by extracellular stimuli is ROS-dependent (Figure 4) [46,47]. For example, exogenous ROS exposure increases both gene expression and protein levels of c-Fos and c-Jun, resulting in a stronger DNA-binding activity in epithelial cells [48]. Thereby, ROS scavengers or antioxidants can effectively inhibit UVB [49,50] or carcinogenic chemical-induced AP-1 activation [51,52]. The DNA-binding activity of c-Fos and c-Jun is determined by the redox state of several conserved cysteine residues [53]. In addition, many studies have revealed that AP-1 is also indirectly regulated by ROS signaling. The AP-1 family of transcription factors are targets for phosphorylation by the mitogen-activated protein kinase (MAPK) cascades, which are ROS responsive, and this leads to enhanced transcriptional activation [45,54]. The precise redox control of AP-1 activation is essential for maintaining cellular homeostasis, and the dysregulation of AP-1 function has been implicated in numerous pathological conditions, including cancer, inflammation and neurodegenerative diseases. 

### 4.2. NF-kB 

The nuclear factor-kappa B (NF-κB) represents a well-known family of transcription factors (NF-κB1, NF-κB2, p65/RelA, c-Rel and RelB) modulating the expression of hundreds of genes involved in cell survival, proliferation, inflammation and immune system function. All of the NF-κB proteins contain a Rel-homology (RHD) domain that is essential for their homo- or heterodimerization and DNA binding. The activation of the canonical NF-κB pathway primarily occurs through the stimulation of proinflammatory receptors, such as the TNF Receptor superfamily, and allows the NF-κB protein dimers to translocate to the nucleus and bind to target genes [55]. Furthermore, NF-κB proteins play pivotal roles in regulation of antioxidative and pro-oxidant genes to protect cells from oxidative stress by ROS (Figure 5) [56]. At moderate concentrations, ROS serve as secondary messengers, actively participating in signal transduction processes that lead to NF-kB activation, whereas many antioxidants effectively block NF-κB activation [57,58]. ROS directly modify and activate the IκB kinase (IKK) complex, which is responsible for phosphorylating the NF-kB inhibitory protein IκB, targeting it for ubiquitin-mediated degradation. This step results in the liberation of NF-kB dimers from their inhibitory complexes, enabling them to translocate into the nucleus and initiate gene transcription [59].

Conversely, when ROS levels surge beyond physiological thresholds, which often occurs in response to pathogens, cytokines or environmental stressors, they can have detrimental effects on NF-kB regulation [60]. Excessive ROS can cause sustained NF-kB activation by impairing the negative feedback mechanisms that normally keep the pathway in check. ROS can directly modify critical cysteine residues within proteins involved in the NF-kB cascade, such as IKK, leading to persistent activation [61]. Furthermore, ROS-induced DNA damage can stimulate NF-kB activation indirectly. The DNA damage sensors, including the ATM and ATR kinases, respond to DNA strand breaks and other lesions by phosphorylating the NF-kB essential modulator (NEMO), a subunit of the IKK complex. This phosphorylation event can promote IKK activation and, consequently, NF-kB activation, linking genotoxic stress to inflammatory responses [62].

### 4.3. p53

p53, often referred to as the “guardian of the genome,” is critical for maintaining genomic integrity and orchestrating cellular responses to stress [63]. ROS intricately modulate the activity of p53, serving as both triggers and regulators within this pathway (Figure 6) [64]. ROS can act as signaling molecules to initiate p53 activation in response to DNA damage. For example, exposure of cells to genotoxic stress such as ionizing radiation or chemotherapeutic agents results in the generation of ROS. This ROS can oxidize specific cysteine residues on the p53 protein leading to a conformational change that stabilizes and activates p53. Once activated, p53 translocates to the nucleus and engages in transcriptional regulation, influencing the expression of genes involved in cell cycle arrest, DNA repair and apoptosis [65]. When ROS levels become chronically elevated, typically due to persistent stressors or pathologies such as chronic inflammation, they can cause severe DNA damage. In this scenario, p53 is activated as a safeguard mechanism to prevent the propagation of damaged cells. It induces cell cycle arrest to allow for DNA repair, and if the damage is irreparable, p53 promotes apoptosis, eliminating cells with compromised genomes [64]. In addition to its direct effects on p53, ROS also impacts on the delicate balance between p53 and its principal negative regulator, MDM2. ROS can oxidize specific cysteine residues within MDM2, disrupting its interaction with p53. This disruption prevents MDM2 from targeting p53 for degradation, resulting in p53 accumulation and elevated activity [66]. 

The interaction between p53 and ROS has been extensively studied [67]. One third of the 48 most highly H_2_O_2_-responsive genes in human cells treated with H_2_O_2_ were identified as p53 targets [68]. A major question is how ROS-regulated p53 results in a differential cell response (e.g., cell cycle arrest, senescence or apoptosis) by selectively regulating certain groups of target genes. It has been proposed that at basal, physiological levels, p53 performs an antioxidant role by maintaining ROS at nontoxic levels through transactivation of antioxidant genes [64]. However, when present at elevated levels, p53 instead can serve as a pro-oxidant by either inducing gene expression, which results in the further elevation of ROS production, or suppressing antioxidant genes. In addition, the p53 protein itself is redox-sensitive due to the presence of the cysteine residues, which can be covalently modified upon redox changes [69]. Therefore, ROS act as pivotal regulators in the p53 signaling pathway, both initiating and fine-tuning p53 activation in response to diverse cellular stressors.

### 4.4. Keap1-Nrf2-ARE

The Keap1-Nrf2-ARE signaling pathway is an extensively studied regulatory system that plays a critical role in preserving cellular redox homeostasis and shielding cells from both internal and external stresses [70]. In this mechanism, ROS act as a central player, providing a dynamic balance between Nrf2 activation and its inhibition by Keap1 (Figure 7) [71].

Under normal conditions, Keap1 serves as a substrate adaptor for the Cullin 3-based E3 ubiquitin ligase complex. It interacts with Nrf2 in the cytoplasm, marking it for ubiquitination and subsequent degradation via the proteasome. This process maintains low levels of Nrf2 in unstressed cells. ROS such as H_2_O_2_ serve as redox messengers that modify this interaction. When cellular ROS levels rise, certain cysteine residues in Keap1 are oxidized. These oxidative modifications cause a conformational change in Keap1, disrupting its ability to ubiquitinate Nrf2. Consequently, Nrf2 accumulates and migrates into the nucleus. Once in the nucleus, Nrf2 forms heterodimers with small Maf proteins, subsequently binding to Antioxidant Response Elements (AREs) situated in the regulatory regions of numerous genes. This interaction initiates the transcriptional activation of an array of antioxidant and detoxification genes, encompassing NADPH quinone oxidoreductase 1 (NQO1), heme oxygenase-1 (HO-1) and GST. These gene products collectively combat oxidative stress and electrophilic insults by neutralizing harmful molecules and enhancing the cellular antioxidant defense system [72].

### 4.5. The MAPK Signaling Pathway

As previously described, the MAPK signaling cascade represents an important link between ROS and AP1 transcriptional regulators (Section 4.1). However, the MAPK signaling cascade plays a more global role in the cell to coordinate multiple transcriptional and non-transcriptional responses to oxidative stress [4]. MAPKs constitute a diverse family of serine/threonine kinases that play a critical role in governing various essential cellular functions such as proliferation, differentiation, stress adaptation and programmed cell death (apoptosis) [73]. The MAPK family is organized into three distinctive subfamilies: the extracellular signal-regulated kinases (ERK) [74], the c-Jun N-terminal kinases (JNK) [75] and the p38 kinases [76]. Each of these subfamilies can be activated independently, although their signaling pathways frequently intersect to transmit signals towards key effector proteins, notably transcription factors that regulate gene expression [73]. The activation of MAP kinases typically involves a cascade of kinase reactions, progressing from MAP kinase kinase kinases (MAPKKKs) to MAP kinase kinases (MAPKKs) and culminating in the activation of the final MAPK effectors [77]. One notable mechanism through which ROS impact MAPK signaling involves the oxidative modification of cysteine residues within key regulatory proteins (Figure 8). ROS-induced oxidation of cysteine residues can activate MAPKKKs, which are upstream components of the MAPK cascade [78,79]. This activation occurs through conformational changes or by facilitating kinase activity. For example, the oxidation of cysteine residues in apoptosis signal-regulating kinase 1 (ASK1), a MAPKKK, leads to its activation. ASK1 then phosphorylates and activates downstream MAPKKs, such as MKK4/7, further propagating the signal [79].

Beside direct activation, ROS also mediate MAPK signaling indirectly, for example by the inhibition of MAPK phosphatases (MKPs). ROS are reported to play a crucial role in inhibiting JNK phosphatases, which are responsible for dephosphorylating and deactivating JNK [80]. The oxidative modification of cysteine residues in MKPs inactivates them, allowing MAPKs to remain phosphorylated and active for a longer period. This sustained MAPK activation is essential for cellular responses to stimuli such as growth factors and stress [81]. 

Intriguingly, MAPKs themselves can generate ROS as part of their signaling. For example, the activation of p38 and ERK MAPKs can induce the expression of ROS-producing enzymes, creating a positive feedback loop [82,83]. This amplifies the ROS signal and reinforces MAPK activation and its downstream effects. In summary, ROS modulate the MAPK signaling pathway by oxidatively modifying key regulatory components, leading to the sustained activation of MAPKs and subsequent cellular responses.

## 5. Role of ROS in DNA Damage Responses

As previously mentioned, DNA is one of the main complex macromolecules that is subject to damage upon oxidative stress and, therefore, the detection and repair of ROS-induced DNA damage is vital for the normal functioning and survival of cells [84]. Left unrepaired, DNA damage can lead to mutations and genomic instability, which are associated with various diseases, including cancer [85]. DNA is vulnerable to numerous lesions like base modifications, single-strand breaks (SSBs) or double-strand breaks (DSBs), ultimately posing a risk of mutations and genomic instability [86]. It has been estimated that oxidative stress can induce approximately 10,000 alterations in DNA per cell per day, encompassing various types of DNA damage. This constitutes a substantial portion of endogenous DNA damage [71,87]. Therefore, amongst the key adaptations for surviving cellular damage induced by ROS is the evolution of effective DNA repair mechanisms that can repair different types of ROS-induced damage and that are temporally coordinated to optimally tackle the damage. Among the well-documented DNA damage types resulting from oxidative stress, the production of 8-Hydroxydeoxyguanosine (8-OHdG), an oxidized derivative of guanine, has received considerable attention. Guanine, owing to its low oxidation potential, is particularly vulnerable to ROS-induced modifications [88].

To counteract these deleterious effects, organisms have evolved a range of DNA repair pathways that can efficiently recognize and rectify different types of DNA damage. For example, the base excision repair (BER) pathway specializes in repairing oxidative damage. In BER, specific DNA glycosylases recognize and remove the damaged bases, creating an abasic or apurinic-apyrimidinic (AP) site. Subsequently, an AP endonuclease cleaves the DNA strand at the AP site, initiating the repair process through the coordinated action of various enzymes and factors, ultimately restoring the original DNA sequence [89]. In this case, 8-OHdG is excised by 8-oxoguanine DNA glycosylase (OGG1) leaving an AP site, followed by either short or long patch BER. The human endonuclease VIII-like 1 (Neil1) protein encoded by the DNA glycosylases Neil-like gene *neil1*, which are homologous to the *E. coli Nei* gene, preferentially eliminates oxidized bases by initiation of base excision repair [90].

Another crucial DNA repair pathway is nucleotide excision repair (NER), which deals with the widest range of structurally unrelated DNA lesions, including ROS-generated cyclopurines, UV-induced pyrimidine dimers and bulky chemical adducts [91]. NER involves a dual incision mechanism that removes a damaged oligonucleotide, followed by resynthesis using the complementary DNA strand as a template. NER incorporates two subpathways: Global Genome NER (GG-NER) and Transcription-Coupled NER (TC-NER). GG-NER is primarily responsible for safeguarding the genome against mutagenesis by actively surveying the DNA for helix-distorting lesions. In contrast, TC-NER is specialized in the removal of lesions that impede the transcription process, thereby ensuring the unimpeded progression of transcription [92]. Within the core excision pathway, the xeroderma pigmentosum complementation group C (XPC), a key protein together with its accessory subunits RAD23 homologue B (RAD23B) and centrin2 (CETN2), serve as the main damage sensor for GG-NER. This complex constantly surveys DNA for helix-distorting lesions with the help of another component, the ultraviolet radiation-DNA damage-binding protein (UV-DDB) complex, which is comprised of the DDB1 and DDB2 proteins. Deficiencies in genes governing NER machinery frequently result in diverse clinical outcomes due to the heightened accumulation of DNA damage. A notable example is observed in individuals afflicted with xeroderma pigmentosum group C, where aberrant XPC protein expression leads to heightened sensitivity to UV radiation and sun-induced skin conditions, substantially increasing the risk of developing skin cancer [93]. In zebrafish, XPC has also been demonstrated to be pivotal in repairing UV-induced DNA damage, since XPC mutant embryos exhibit significantly higher levels of DNA damage and apoptotic cells upon exposure to UV radiation, leading to severely impaired development [94]. Thus, XPC is an essential component of the NER system for DNA damage recognition in vertebrates and, consistent with this role, the expression levels of both *xpc* and *ddb2* are robustly induced by visible light, UV and ROS, which are proxies for sunlight exposure. 

Many repair systems which target damage induced by ROS are themselves activated by ROS [95]. Numerous studies in mammals have demonstrated that DNA repair genes are regulated by cellular ROS indirectly via the activation of key transcription factors involved in DNA repair signaling. For instance, this includes the activation of transcription of the repair genes *apex1* and *neil1* in BER that occurs through the AP-1 pathway [96]. ROS-activated CREB/c-Jun has been shown to bind to the AP-1 site in the *neil1* promoter and thereby to up-regulate mRNA expression of *neil1* in mammalian cells [97]. Furthermore, a study of changes in the zebrafish transcriptome in response to ROS has revealed that similar gene expression programs are shared by zebrafish and human cells, indicating a generally conserved transcriptional regulation effect of ROS [98]. However, direct transcriptional regulation by ROS has been studied in the case of only a few DNA repair genes and detailed mechanisms have not been well elucidated. A more detailed understanding of the ROS-mediated transcription of DNA repair genes should provide a better understanding of their essential role in physiological responses.

## 6. Fish as Models to Study How ROS Contribute to Physiological Systems

The choice of animal model is of fundamental importance for defining which facets of ROS function and regulation can be effectively studied. Fish, constituting the largest group of vertebrates, hold significant ecological importance and substantial commercial value. As a commonly used fish model, the zebrafish (*Danio rerio*) represents a versatile and valuable tool for exploring various genetic and molecular mechanisms which underlie behavior, physiology and cell biology. Its optical transparency during embryonic and larval stages allows for non-invasive and real-time visualization of the dynamics of gene expression within a living organism [99]. Within the context of this review, studies on zebrafish have revealed many physiological roles of ROS in various biological processes. 

In cardiovascular development, ROS play a pivotal role in angiogenesis, cardiomyocyte proliferation and tissue regeneration. For example, experiments using zebrafish have revealed the critical role of the HECT domain and Ankyrin repeat-containing E3 ubiquitin-protein ligase 1 (hace1) in the normal development and function of the vertebrate heart in an ROS-dependent manner [100]. Expression of hace1 negatively regulates NOX-dependent ROS generation to maintain normal cardiac development, whereas knockdown of hace1 results in ROS accumulation and cardiac defects. Within the realm of neurobiology, zebrafish have provided insight into the contribution of ROS to neural development and function as well as axonal regeneration. Specifically, H_2_O_2_ promotes peripheral sensory axon growth in the skin, which is crucial for cutaneous injury healing [101]. Another study has reported that caudal fin amputation in adult zebrafish results in H_2_O_2_ production in the wounded epidermis, and then activation of hedgehog signaling, probably by the transcriptional activation of the sonic hedgehog gene (Shh) [102]. Taking advantage of the use of ROS-specific biosensors and the optical transparency of zebrafish embryos and larvae, ROS generation upon wounding has been visualized and quantified. However, the detailed mechanism for how ROS are generated and how in turn ROS regulate down-stream signaling has not been completely elucidated. Therefore, zebrafish studies have shed light on how ROS are involved in various important physiological and pathological processes and provide an important model for studying various human diseases. Notable mechanisms where ROS play a key regulatory role include the circadian clock, as well as DNA repair mechanisms and recently, these interconnections have been studied extensively in various fish models including zebrafish and blind cavefish.

### 6.1. Links between ROS, DNA Repair and the Circadian Clock

Given the links between sunlight exposure and general levels of cellular metabolic activity with the generation of ROS (Figure 1), it is evident that levels of oxidative stress as well as associated macromolecular damage tend to vary significantly between the day and night. Therefore, in turn, the regulation and activity of ROS-responsive mechanisms including DNA damage repair are far from constant over the course of the day-night cycle and are closely linked with the function of the circadian clock. Indeed, daily fluctuations in oxidative stress are considered to have served as a significant selective pressure underlying the evolution of the circadian clock. This adaptation allowed organisms to anticipate variations in ROS levels associated with the extended exposure of cells and tissues to sunlight before their actual occurrence, thus facilitating the optimal coordination of repair and survival strategies [103]. 

The circadian clock is a timing mechanism that plays a key coordinating role in synchronizing the physiology of organisms with the day-night cycle [104]. The clock is a cell-autonomous and self-sustaining mechanism that is present in most tissues and cell types and operates independently of external stimuli. However, it is reset on a daily basis by environmental signals which are indicative of the time of day, so-called “zeitgebers” (time-givers), primarily light. In mammals, peripheral organs like the heart, liver and skin possess cell-autonomous clocks that are synchronized through neural and humoral signals stemming from the central “master clock” located within the brain [105]. At the molecular level, the mechanism underlying the circadian clock relies on a cell-autonomous transcriptional autoregulatory feedback loop. The key constituents of this core clock machinery consist of the activator transcription factors CLOCK and BMAL1, along with the transcriptional repressors PERIOD (PER) and CRYPTOCHROME (CRY) [106]. In recent years, ROS-responsive genes have been found to exhibit time-of-day-specific changes of expression both in plants [107] and animals [108,109]. ROS appear to function both as an input signal and a target for intracellular clock function in mammals, which could further influence circadian clock-controlled downstream transcriptional responses. For example, endogenous ROS levels were found to oscillate rhythmically in mammalian cells, which in turn regulate circadian clocks by the redox control of the CLOCK protein [108]. In addition, the expression level of antioxidant enzymes is regulated primarily by the transcription factor Nrf2, which is under the circadian control by BMAL1: CLOCK complex [110,111]. Thereby, the antioxidative systems which regulate intracellular ROS levels are clock-regulated. Therefore, the redox state of cells and the circadian clock are tightly interconnected.

Studies using zebrafish have revealed that fundamental differences in the function and organization of photic responses, DNA repair and the circadian clock exist between fish and other major vertebrate groups. In mammals, the circadian timing system consists of a set of peripheral clocks located in most organs and tissues that are coordinated by specialized “central pacemakers”, notably the suprachiasmatic nucleus, and which rely upon light detection by non-visual photoreceptors in the retina for entrainment by light. In contrast, all peripheral tissue clocks in fish are entrainable via direct light exposure, a property which is even shared by fish-derived cell cultures. A surprisingly diverse set of non-visual opsin photoreceptors is widely expressed in most fish cell types and tissues, and importantly visible light exposure triggers the transcription of a set of genes which includes clock genes, genes related to DNA damage repair (including the photolyase genes *cpd*, *6-4phr* and *cry-dash* and the NER gene *ddb2*) as well as genes involved in various aspects of metabolism. Light-induced transcription is directed by D-box enhancers and the PAR-E4BP4 family of transcription factors. Interestingly, the D-box enhancer and its associated transcription factors also mediate UV and ROS-induced transcription. This situation differs considerably from the role of the D-box in mammals, where D-box-regulating transcription factors are clock-regulated and, therefore, the D-box enhancer serves as a clock output mechanism [42]. This points to certain key elements of ROS-mediated transcription control being relatively plastic over the course of evolution. 

### 6.2. Role of the bZIP PAR/E4BP4 Factors in ROS- and Light-Regulated Transcription

The PAR-domain and E4BP4 factors are bZIP transcription factors which are highly conserved in animals and have been demonstrated to participate in the core circadian clock feedback loops. The three PAR factors, thyrotroph embryonic factor (TEF), hepatic leukemia factor (HLF) and albumin D-site-binding protein (DBP), serve as transcriptional activators, while the E4 binding protein 4 (E4BP4) has been shown to act as a repressor. They share a conserved bZip-DNA-binding-dimerization domain but in the case of PAR factors, they also contain a conserved proline and acidic amino acid-rich (PAR) domain [112]. In certain animal groups such as fish, multiple homologs of PAR factors and E4BP4 have been identified, namely TEF1, TEF2, HLF1, HLF2, DBP1, DPB2 and six members of E4BP4 (E4BP4-1 to 6) [113]. 

In mammals, it has been demonstrated that PAR/E4BP4 gene expression and function are directly regulated by the core clock machinery. Specifically, the rhythmic expression of DBP is driven by the CLOCK:BMAL complex through E-box-mediated activation [114]. In turn, these clock-controlled transcription factors impart circadian rhythmicity on downstream genes, thereby influencing diverse physiological processes, including hepatic xenobiotic metabolism and detoxification [113]. A fundamentally different role for PAR factors in transcriptional regulation has been identified in zebrafish. A previous study has demonstrated that TEF1 activates the *per2* promoter by binding to the D-box enhancer element in zebrafish in response to light, indicating a novel clock input function [115]. Further research has revealed that light-induced transcription that is mediated by the D-box is also ROS-dependent [116]. This demonstrates that PAR/E4BP4 factors serve as key players in the transcriptional response to ROS in vertebrates, and also provides evidence that this functionality may have been adapted significantly over the course of evolution.

### 6.3. Evolutionary Perspectives on ROS-Mediated Transcriptional Regulation in Fish

While the fundamental principles of ROS-mediated transcriptional regulation appear to be conserved in animals, there are still many questions which remain incompletely answered. Our understanding of mechanisms aiding organisms to survive elevated ROS levels has largely stemmed from studies using a limited number of genetic and cell culture models, predominantly of mouse and human origin. Therefore, our knowledge about the conservation or adaptation of these mechanisms throughout vertebrate evolution under diverse environmental conditions remains relatively sparse. Thus, for example, are the gene regulatory pathways that respond to increased ROS levels identical in all vertebrate groups? Does evolution under environmental conditions, where there are significant differences in the levels of oxidative stress, result in alterations to these basic mechanisms? Answers to these basic questions are vital for a more general understanding of the mechanisms whereby toxic compounds and the environment impact on a range of different organisms and how this process has shaped the evolution of redox signaling and antioxidation mechanisms.

Clues as to the evolutionary selection pressures that may have shaped ROS-responsive transcription control mechanisms in vertebrates have come from comparative studies using species of cavefish. The biology of various subterranean species which have evolved in complete isolation from sunlight has provided us with valuable and unique insight into how organisms evolve in response to extreme environments, characterized by constant darkness and temperature and limited food availability. Both terrestrial and aquatic species which have evolved in these conditions show common features known as “troglomorphisms”, including degenerated visual systems, starvation tolerance, increased longevity and, in particular, the loss of body pigmentation and eyes [117,118,119]. In contrast, their sensory systems exhibit enhanced sensitivity for the detection of chemical and mechanical stimuli in the absence of visual navigation clues. These adaptations are crucial for foraging and navigating within the cave environment, which is characterized by limited food resources and intricate subterranean landscapes [120,121]. 

There are more than 200 blind fish species that have been identified living in different cave environments, and the Mexican tetra (*Astyanax mexicanus*) is one of the more extensively studied species. They have both eyed surface forms, which are widely distributed in northeast Mexico and south Texas, and several eyeless cave populations—for example, a typical population called “Pachón”. Crosses between these two forms generate fertile offspring and, therefore, this species can be used to explore the genetics which underlies troglomorphisms as well as providing an advantageous and comparative model for the study of evolutionary genetics [122]. Studies on cave populations of *A. mexicanus* indicated a retaining but altered circadian oscillation compared with surface fish, due to increased basal levels of light-inducible genes such as *per2* [123].

The Somalian cavefish, *Phreatichthys andruzzii*, inhabits subterranean waters beneath the central Somalian desert. Studies have demonstrated that this species initially colonized cave habitats five million years ago and has been completely isolated from surface waters for about three million years. Therefore, compared with *A. mexicanus*, *P. andruzzii* has been adapting to its cave environment for a significantly longer time period and, as a result, exhibits a much stronger troglomorphic phenotype. For example, *P. andruzzii* exhibits complete loss of body pigmentation and the visual system including loss of the eyes, optic nerves and chiasma [124]. Importantly, this was one of the first animals discovered with a dysfunctional biological clock that was no longer entrained by the light/dark cycle [125]. 

Both zebrafish and *P. andruzzi* belong to the *Cyprinidae* family, and thereby share significant similarities in their genetic makeup. With their close genetic relationship, the comparison of the genomes or specific genes of *P. andruzzii* and zebrafish can provide powerful insight into the specific genetic variation and adaptations that have occurred during evolution in its cave environment. Many genetic tools and approaches established in zebrafish can also be applied to *P. andruzzii*. So far, detailed molecular characterization of the circadian clock and DNA repair systems on this fish model has been performed—for example, the loss of light inducibility of clock genes [125] and the loss of photoreactivation DNA repair [95]. Previous studies demonstrated that light-induced gene expression in fish cell lines is determined by the generation of intracellular ROS levels [95,116]. Considering that cavefish evolved over prolonged periods in extreme environments characterized by perpetual darkness and hypoxic aquatic environments, alteration of ROS signaling pathways might have been a causative factor contributing to the loss of light/clock-regulated transcriptional responses.

Previous work has shown striking differences in the transcriptional regulatory mechanisms which respond to light, UV and ROS between zebrafish and the Somalian cavefish (*P. andruzzii*). The D-box enhancer element appears to mediate light and UV-induced transcriptional responses in clock and DNA repair genes in zebrafish. Upon exposure to light and UV radiation, ROS levels increase in zebrafish cells, leading to the activation of the p38 and JNK MAPK pathways. Consequently, these MAPKs trigger the activation of PAR transcription factors, which bind to the D-box, initiating the transcription of a specific set of clock and DNA repair genes. Intriguingly, this D-box-mediated transcriptional process appears significantly attenuated in Somalian cavefish, (Figure 9) [42,95]. As previously described, in contrast the D-box element in mammals serves as a component of the circadian network that involves the CLOCK/BMAL1, PER/CRY and REV-ERB/RORs interlocking transcriptional feedback loops [106]. Similar to cavefish, no increase in D-box-driven transcription is observed in mammalian cells upon H_2_O_2_ exposure [116]. These differences between zebrafish, cavefish and mammals point to a degree of plasticity in transcription control of D-box element regulation during evolution in different environments. Therefore, studies of different transcriptional responses to ROS with an evolutionary perspective are of fundamental importance.

## 7. Conservation and Evolution of ROS Signaling

From a broader perspective, how has the evolution of ROS responsive mechanisms been shaped by the redox state of the environment? The very earliest forms of life survived the earth’s atmosphere, which was dominated by volcanic gases and mainly comprised hydrogen, carbon dioxide, carbon monoxide, hydrogen sulfide and methane [126]. Under this “toxic” environment, it is predicted that the first forms of life, the archaea and bacteria, emerged probably in alkaline thermal vents in the oceans [127]. Thereafter, the advent of cyanobacteria began converting water to the hydrogen and oxygen required for metabolic reactions, marking the origins of the aerobic atmosphere and the explosion of life on earth. Since the dramatic rise of atmospheric O_2_ during the Precambrian period, an expansion of multicellular species occurred, facilitating the evolution of complex life [128]. With elevated O_2_ levels, the consumption rate of O_2_ was coupled with complex mechanisms which ensured ROS homeostasis and led to ROS being recruited to serve in signaling networks of organisms. 

Considering the general mechanism of ROS production in cells, which is ubiquitous in aerobic organisms, and the common role of ROS in the regulation of cell metabolism, development and responses to the environment, ROS signaling in response to endogenous and exogenous stimuli is regarded as being conserved across species from prokaryotes to eukaryotes [129]. Indeed, a number of studies have demonstrated similar mechanisms of ROS homeostasis being involved in various species. For example, the thioredoxin (TXR) families play a pivotal role in catalyzing oxidoreductase reactions aimed at reducing disulfide bonds within specific target proteins. These enzymes are recognized as potent reductants and are ubiquitously present across various life forms, including bacteria, fungi, plants and mammals [130]. Notably, akin to yeast and bacterial systems, the functional significance of TRX is underscored in vertebrates, as exemplified by the observation of embryo lethality in mice upon the loss of TRX1 [131]. GRXs are indicated as essential for plant development with the double mutant of *grxc1* and *grxc2*, leading to embryo lethality [132]. Glutaredoxin (GRX) proteins exhibit a pivotal function as essential redox transmitters within the thiol/disulfide redox network, modulating a myriad of cellular processes, notably development. Additionally, a diverse array of ROS-related proteins, including thiol peroxidases known for their high peroxide affinity in shielding target protein thiols from oxidation, and the NADPH oxidase family responsible for generating superoxide radicals by harnessing NADPH as an electron donor, appear ubiquitously across various kingdoms. These proteins collectively contribute significantly to developmental processes by sensing ROS and maintaining crucial redox equilibrium within cells [129]. 

In contrast to these evolutionarily ancient and conserved mechanisms of ROS signaling, a diversity of key redox signaling regulators has been documented. For example, the SOD and NOX enzyme families comprise some conserved members but also notable variations across phyletic animal lineages. SOD1 and SOD2 are widely distributed across the metazoans, whereas SOD3 is absent in most sponges, a group of species which occupy a unique phylogenetic position as sister to other animal phyla. Traits unique to sponges have been logically traced back to the divergent evolution from a common ancestor [133]. A study of the ctenophore *Mnemiopsis leidyi* demonstrated substantial NOX gene loss with the retention of only NOX5 [133,134]. Ctenophores have been considered the earliest diverging animal lineage which share with bilaterians complex cell types such as neural cells [135]. *M. leidyi* exhibits daily vertical migration in response to high radiation levels, which may affect their cellular redox state. Indeed, a comparatively wide range of Cu/ZnSODs are encoded by *M. leidyi*, which has been suggested to compensate for the reduction in the diversity of encoded NOX enzymes [133]. 

Nrf2-Keap1 signaling is identified as the major regulator of oxidative stress response and considered evolutionarily conserved in metazoans [136]. A recent study regarding the Keap1 protein, which is a crucial component of the Nrf2-Keap1 signaling pathway, revealed that the molecular evolution of Keap1 from lower to higher vertebrates has been indispensable for adaption to terrestrial life [137]. The Keap1A gene, a zebrafish paralog which is absent in mammals, showed a stronger affinity for Cul3-RING ubiquitin ligase-mediated degradation, resulting in a lower Nrf2-mediated antioxidation compared to terrestrial species. This may explain how vertebrates successfully adapted to terrestrial conditions where they encountered a higher level of oxidative stress compared to the aquatic environment. 

In *Drosophila*, the Cap’n’collar (*CncC*) gene encodes the CncC protein, which is homologous to mammalian Nrf2 [138]. The evolutionary conservation between the *CncC* gene in *Drosophila* and the Nrf2 pathway in mammals provides valuable insight into the conservation of cellular stress response mechanisms. This conservation extends to other key components such as Keap1 and Maf, underscoring the pathway’s fundamental importance across species. The heightened sensitivity to oxidative stress observed in Nrf2-deficient mice, without resulting in lethality, suggests the presence of compensatory mechanisms, possibly involving other members of the Nrf family, such as Nrf1 or Nrf3 [139,140]. In contrast, the developmental lethality seen in *Drosophila* lacking CncC underscores its essential role, which likely extends beyond antioxidation, reminiscent of the multifaceted functions of Nrf1 [141]. The proposition that CncC in insects may represent an ancestral form of both Nrf1 and Nrf2 in mammals is intriguing, indicating potential evolutionary diversification in the functions of these transcription factors [138]. This implies that insects such as *Drosophila* may retain a more ancestral, multifunctional version of the protein.

Another example is the naked mole-rat, *Heterocephalus glabera*, a representative rodent which lives an underground dwelling lifestyle. With strong hypoxia tolerance and high DNA repair capacity, naked mole-rats tend to have the longest lifespan (maximum 37 years) among rodents and many fewer aging-related diseases [142]. Several studies have reported that individuals of *H. glabera* are resistant to ROS and pro-oxidant toxins [143,144]. Higher consumption of H_2_O_2_ in the mitochondria of *H. glabera* tissues [145] and the highly activated Nrf2-Keap1 signaling pathway [146] may contribute to the robust ROS resistance in this species. Comparable to cavefish, the biological and molecular characteristics of the naked mole-rat have been profoundly influenced by evolution and its environment, contributing to exceptional longevity, tolerance to hypoxia and resistance to carcinogenesis. The strategies developed by these organisms to respond to ROS and oxidative stress are believed to be pivotal in their adaptation to subterranean environments.

Overall, the ROS signaling networks observed in diverse species appear to be tightly connected with the ecological niches which they occupy, and the diversity of signaling elements could provide new sight into the early evolution of key molecular mechanisms which combat environmental oxidative stress.

## 8. ROS in Human Disease

Dysregulation of ROS signaling is tightly interconnected with human disease. For example, oxidative stress is a fundamental characteristic of neurodegeneration and significantly contributes to the advancement of neuronal damage in Alzheimer’s Disease (AD). This damage to neurons mediated by ROS arises from disturbances in redox reactions, marked by declines in the activities of antioxidant enzymes including SOD and CAT, as well as reductions in the levels of antioxidants such as ascorbic acid and tocopherol [147]. This imbalance leads to elevated steady-state levels of ROS, exacerbating cellular damage. Moreover, the accumulation of amyloid-beta (Aβ) in individuals with AD further intensifies oxidative stress, constituting a cycle that worsens neuronal dysfunction and hastens disease progression [148]. ROS also play a multifaceted role in cancer initiation and progression, contributing to the complex landscape of carcinogenesis [149]. Excessive ROS production induces oxidative stress, leading to DNA damage, lipid peroxidation, and protein modifications, all implicated in oncogenic transformation. ROS-mediated activation of redox-sensitive signaling pathways, such as nuclear factor kappa B (NF-κB) and mitogen-activated protein kinase (MAPK), promotes tumor cell proliferation, survival, angiogenesis and metastasis [150]. Conversely, cancer cells often exhibit enhanced antioxidant defenses to counteract ROS-induced damage, conferring a selective advantage for survival and tumor progression [151].

Since ROS play pivotal roles in pathogenesis, several therapeutic strategies targeting ROS hold promise for various diseases, including cancer, by exploiting the delicate redox balance within cancer cells [152]. One such approach involves photodynamic therapy, which harnesses the generation of ROS through the stimulation of photosensitizers by light [153]. This results in the selective induction of oxidative stress within cancer cells, leading to their demise while sparing healthy tissues. In addition, modulation of antioxidant enzymes represents another potential therapeutic strategy, as inhibition of these enzymes can sensitize cancer cells to treatments that increase ROS levels, thereby enhancing therapeutic efficacy [154]. Furthermore, studies have shown that depleting ATP, whether through the manipulation of glycolytic enzymes, chemotherapy or radiation therapy, can induce ROS-mediated apoptosis in cancer cells, highlighting the potential of combinatorial therapies targeting ROS [155]. However, recent research has challenged the traditional notion of using antioxidant drugs in cancer therapy, instead suggesting that their administration may promote tumor progression. Therefore, a shift towards inhibiting antioxidant systems in combination with ROS-inducing treatments represents a promising alternative [152].

Currently, one significant challenge is achieving specificity in ROS modulation to selectively target cancer cells while sparing healthy tissues. ROS are involved in numerous physiological processes, and indiscriminate manipulation can lead to off-target effects and unintended consequences, including cytotoxicity in normal cells and tissues [156]. In addition, the complex interplay between ROS and various signaling pathways necessitates a comprehensive understanding of ROS dynamics within the tumor microenvironment in order to develop effective therapeutic strategies. For example, the dual role of ROS as both mediators of cell death and promoters of cell survival complicates therapeutic targeting. Depending on the context and cellular conditions, ROS can induce apoptosis or promote cell proliferation and metastasis [152]. Thus, fine-tuning ROS levels to achieve the desired therapeutic outcome while avoiding unintended consequences poses a significant challenge. As a step towards this goal, the emergence of nanomedicine has revolutionized cancer therapy by enabling targeted approaches that capitalize on the unique properties of cancer cells [156]. This innovation allows for the delivery of pro-oxidant agents specifically to primary lesions and subcellular structures within tumors, leading to significant success in eliminating cancer cells while minimizing harm to healthy tissues [157,158]. While antioxidant therapy alone has shown limited efficacy in clinical cancer treatment, the combination of antioxidant agents with chemoradiotherapy has demonstrated remarkable outcomes in clinical trials. However, more collaborative efforts involving basic research, preclinical studies and clinical trials will be needed to realize the full potential of ROS modulation in the treatment of various kinds of disease.

## 9. Conclusions and Perspectives

ROS play a multifaceted role in cell physiology, acting as a “double-edged” sword. On one hand, while ROS can inflict substantial damage to macromolecules, their excessive accumulation can trigger oxidative stress, contributing to the onset of various diseases. On the other hand, ROS also function as crucial signaling molecules involved in numerous cellular activities, such as redox homeostasis, circadian clock entrainment, gene expression regulation, immune response and DNA repair. Currently, our understanding of the ROS-mediated signal transduction and transcriptional regulation demonstrate a remarkable conservation of many signaling pathways among vertebrates. However, some fundamental differences have been revealed from fish to mammals—for example, the ROS-responsive D-box-mediated transcription we discussed in this review. Comparative studies involving diverse vertebrates, including zebrafish and cave-dwelling species like *P. andruzzii*, have provided valuable new perspectives on the adaptive alterations in ROS-associated pathways. These investigations emphasize the delicate balance between conserved elements and adaptive modifications in ROS-mediated signaling, shedding light on how different species have fine-tuned responses to oxidative stress based on their ecological contexts. 

Given these findings, it is evident that a comprehensive understanding of the biology of ROS should fundamentally account for how they can generate macromolecular damage and, at the same time, also serve as key cell signaling molecules. However, it should also explain how each species adapts to the unique profile of oxidative stress that over the course of evolution it encounters in its environment and as a result of its particular lifestyle. Therefore, moving forward, further studies should delve into broader comparative genomics and functional studies across a wider array of species, to unravel the diverse nuances and evolutionary trajectories of ROS-related mechanisms. Furthermore, they should clarify the interplay between ROS and epigenetics as a potential mechanism accounting for evolutionary change. Importantly, this approach should not only allow us to elucidate the underlying mechanisms whereby ROS directly contribute to pathogenesis or immune defenses, but also provide a global and precise view of the species-specific “targets” of ROS. These could potentially serve as novel molecular targets for antioxidants or genetic modifications in new therapeutic strategies to counter ROS-related diseases, such as aging, cancer, neurodegeneration and chronic chemical intoxication.

## Figures and Tables

**Figure 1 antioxidants-13-00312-f001:**
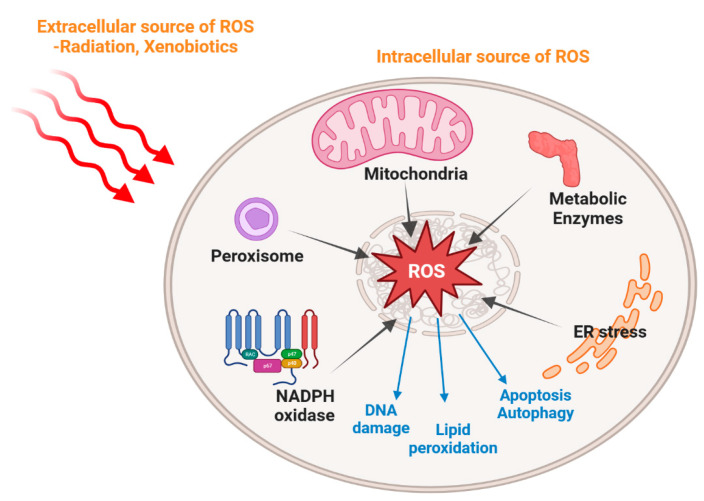
Schematic representation of oxidative stress induced by intracellular and extracellular stimuli. Extracellular sources of ROS include environmental factors such as radiation and xenobiotics. Intracellular ROS are generated mainly in the mitochondria, as well as other organelles such as the peroxisome and endoplasmic reticulum, and in addition by some metabolic enzymes. Excessive ROS production can lead to cell apoptosis, autophagy, lipid peroxidation and DNA damage.

**Figure 2 antioxidants-13-00312-f002:**
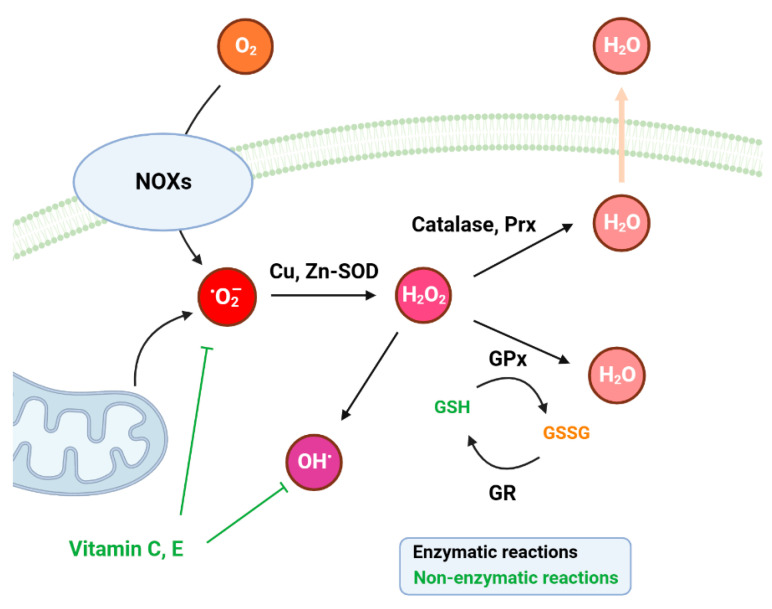
Schematic presentation of antioxidation systems with enzymatic and non-enzymatic antioxidants. Black arrows indicate enzymatic reactions and green lines indicate non-enzymatic reactions. NOXs: the NADPH oxidases; SOD: superoxide dismutase; Prx: peroxiredoxin; GPx: glutathione peroxidase; GR: glutathione reductase; GSH: glutathione; GSSG: glutathione disulfide.

**Figure 3 antioxidants-13-00312-f003:**
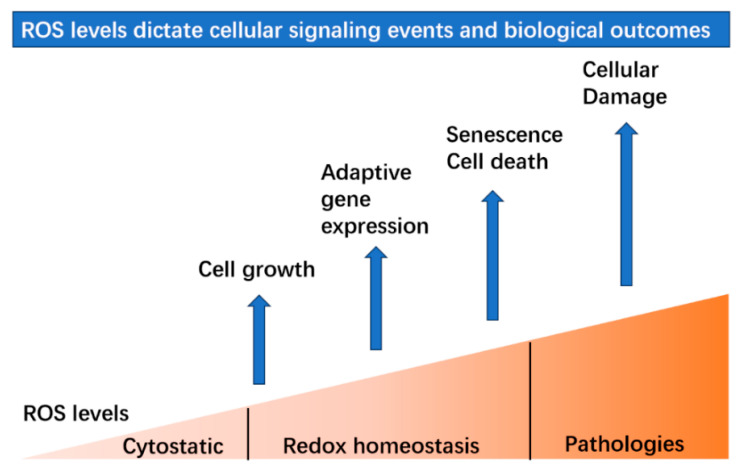
ROS levels dictate different signaling events and biological outcomes. Within the range of normal physiological levels, ROS participate in many important cellular processes to maintain redox homeostasis. At low levels, ROS promote cell growth by proliferation and differentiation. The induction of ROS levels leads to adaptive gene expression such as the up-regulated expression of antioxidative genes. Then, exposure to higher levels will result in the initiation of senescence or cell death.

**Figure 4 antioxidants-13-00312-f004:**
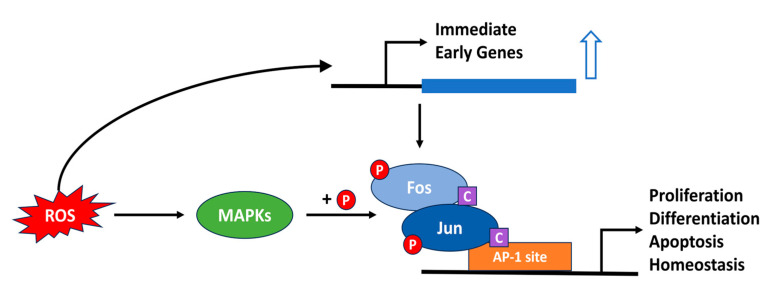
ROS-mediated regulation of the AP-1 transcription factor family. ROS act as signaling molecules, activating AP-1 protein complexes and enhancing their transcriptional activation function. A key step in this process is the phosphorylation of the AP-1 factors by MAPKs (+P). In addition, changes in the redox state of certain conserved cysteine residues (C) have been implicated in regulating DNA binding. Subsequently, the expression of genes involved in fundamental cellular processes is regulated. Excessive amounts of ROS lead to aberrant AP-1 activation, potentially disrupting cellular homeostasis.

**Figure 5 antioxidants-13-00312-f005:**
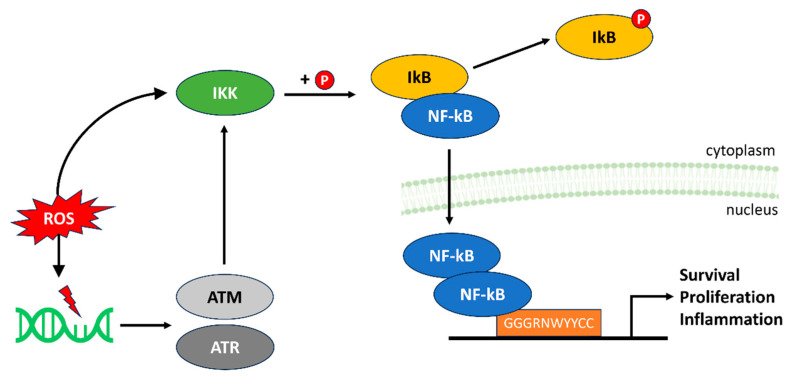
ROS modulation of the NF-κB pathway. At moderate levels, ROS facilitate NF-κB activation and translocation to the nucleus to initiate gene transcription. This involves the phosphorylation of IkB by IKK (indicated by P), thereby targeting it for ubiquitin-mediated degradation and liberating NF-kB dimers, which translocate to the nucleus. Excessive ROS levels directly and indirectly disrupt IKK regulation, resulting in sustained activation of NF-κB and inflammatory responses.

**Figure 6 antioxidants-13-00312-f006:**
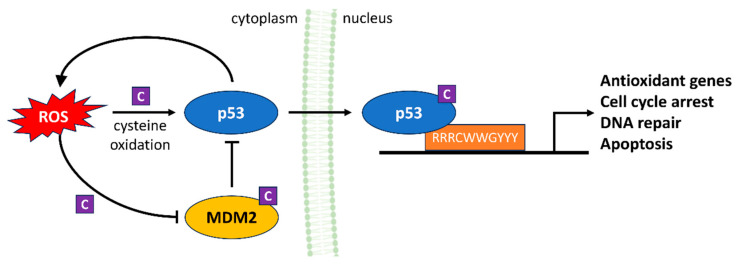
ROS regulation of the p53 signaling pathway. ROS serve as both triggers and regulators of p53 activation in response to cellular stressors. At physiological concentrations, ROS stabilize and activate p53, promoting DNA repair and antioxidant responses, preventing spreading of damaged cells and keeping ROS at non-toxic levels. These effects of ROS on p53 as well as its regulatory factor MDM2 are mediated by oxidation of cysteine residues in both proteins (C). However, high p53 levels lead to pro-oxidant responses, increasing ROS and suppressing antioxidant genes.

**Figure 7 antioxidants-13-00312-f007:**
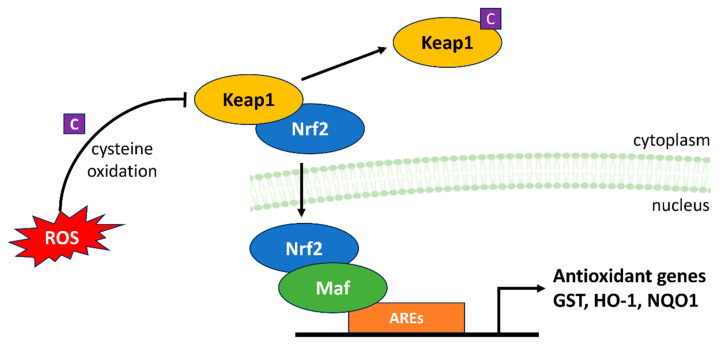
ROS modulation of the Keap1-Nrf2-ARE signaling pathway. ROS play a central role in the balance between Nrf2 activation and inhibition. At physiological levels, Keap1 maintains Nrf2 at low concentrations. However, at high ROS concentrations, Keap1 is oxidized at particular cysteine residues (C) and Nrf2 degradation is prevented, thereby increasing transcription of antioxidant genes via AREs and promoting defense against oxidative stress.

**Figure 8 antioxidants-13-00312-f008:**
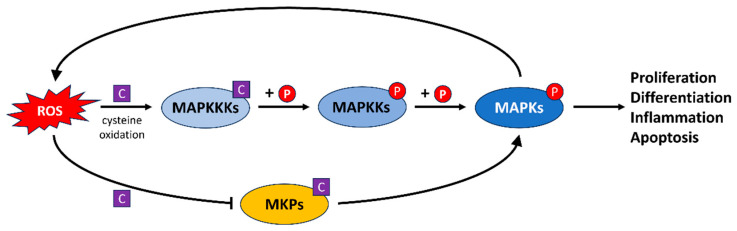
ROS regulation of the MAPK signaling pathway. ROS serve as crucial regulators of the MAPK signaling pathway both directly by activating upstream kinases, and so enabling them to phosphorylate their downstream targets (P), and indirectly by inhibiting MAPK phosphatases via the oxidation of cysteine residues (C) on these enzymes. MAPKs can increase ROS levels, reinforcing the signal and its downstream effects.

**Figure 9 antioxidants-13-00312-f009:**
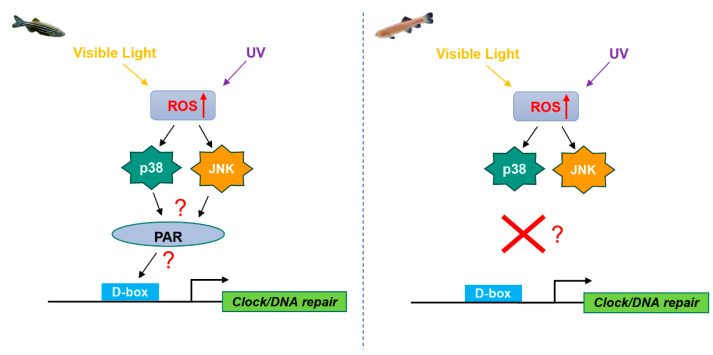
Schematic representation of the ROS response of the D-box element in zebrafish and cavefish. In zebrafish cells (**left panel**), elevated levels of ROS are induced by exposure to both visible and UV light. Then, ROS activate MAPKs including p38 and JNK rapidly by phosphorylation and in turn activate the PAR factors. PAR factors bind to D-box enhancer elements in the promoters of a set of clock and DNA repair genes and ultimately lead to induced gene transcription and clock entrainment. In cavefish cells (**right panel**), this signaling fails to activate transcription of the same genes via the D-box [95].

## Data Availability

No new data were created in the current study.

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
