# Peer review of "Reactive Oxygen Species Signaling and Oxidative Stress: Transcriptional Regulation and Evolution"

_antioxidants, 2024, doi:10.3390/antiox13030312_

Round 1

Reviewer 1 Report

1. Your manuscript provides a thorough and detailed overview of ROS (Reactive Oxygen Species) and their diverse roles in biology. The sections are well-organized, covering topics from basic biochemical aspects to evolutionary perspectives. However, it would be beneficial to include a brief introductory summary at the beginning of each section to guide the reader through the complex content.

2. While the manuscript mentions figures (e.g., Fig 1, Fig 2), including detailed, clear, and well-labeled diagrams or flowcharts could significantly enhance the reader's understanding, especially of complex pathways and mechanisms.

3. The manuscript discusses various aspects of ROS in general terms. Including more specific case studies or examples, particularly in sections discussing the role of ROS in diseases or cellular mechanisms, would provide a more concrete understanding of these processes.

4. Ensure that the manuscript includes the latest research findings and technological advancements in the field of ROS research. This might involve discussing new methods for detecting and measuring ROS, recent discoveries in ROS signaling pathways, or novel therapeutic strategies targeting ROS.

5. The evolutionary perspective is intriguing, especially the discussion about fish models. Expanding this section to include a comparative analysis with other model organisms (e.g., mice, fruit flies) could provide a more comprehensive view of ROS mechanisms across different species.

6. The manuscript touches on the implications of ROS in various diseases. A more detailed discussion on current and potential therapeutic strategies targeting ROS, including antioxidants and gene therapy, could be highly valuable. This section could also discuss the challenges and limitations of these therapeutic approaches.

7. Concluding the manuscript with a section on future research directions would be beneficial. Highlighting unanswered questions and proposing potential research areas could spur further investigation in the field of ROS. This might include emerging fields like the interplay between ROS and epigenetics, or the impact of environmental factors on ROS generation and regulation.

Overall, your manuscript is informative and covers a broad range of topics related to ROS. These suggestions aim to further enhance its comprehensiveness and utility to readers in the field.

1. Your manuscript provides a thorough and detailed overview of ROS (Reactive Oxygen Species) and their diverse roles in biology. The sections are well-organized, covering topics from basic biochemical aspects to evolutionary perspectives. However, it would be beneficial to include a brief introductory summary at the beginning of each section to guide the reader through the complex content.

2. While the manuscript mentions figures (e.g., Fig 1, Fig 2), including detailed, clear, and well-labeled diagrams or flowcharts could significantly enhance the reader's understanding, especially of complex pathways and mechanisms.

3. The manuscript discusses various aspects of ROS in general terms. Including more specific case studies or examples, particularly in sections discussing the role of ROS in diseases or cellular mechanisms, would provide a more concrete understanding of these processes.

4. Ensure that the manuscript includes the latest research findings and technological advancements in the field of ROS research. This might involve discussing new methods for detecting and measuring ROS, recent discoveries in ROS signaling pathways, or novel therapeutic strategies targeting ROS.

5. The evolutionary perspective is intriguing, especially the discussion about fish models. Expanding this section to include a comparative analysis with other model organisms (e.g., mice, fruit flies) could provide a more comprehensive view of ROS mechanisms across different species.

6. The manuscript touches on the implications of ROS in various diseases. A more detailed discussion on current and potential therapeutic strategies targeting ROS, including antioxidants and gene therapy, could be highly valuable. This section could also discuss the challenges and limitations of these therapeutic approaches.

7. Concluding the manuscript with a section on future research directions would be beneficial. Highlighting unanswered questions and proposing potential research areas could spur further investigation in the field of ROS. This might include emerging fields like the interplay between ROS and epigenetics, or the impact of environmental factors on ROS generation and regulation.

Overall, your manuscript is informative and covers a broad range of topics related to ROS. These suggestions aim to further enhance its comprehensiveness and utility to readers in the field.

Author Response

We would like to thank this reviewer for their very positive and constructive comments. We agree entirely with all the points that they have raised and have adapted the text accordingly.

1…. It would be beneficial to include a brief introductory summary at the beginning of each section to guide the reader through the complex content.

We agree and throughout our manuscript, we have now included additional sentences which aim to more effectively bridge different parts of the text.

  1. …..including detailed, clear, and well-labeled diagrams or flowcharts could significantly enhance the reader's understanding, especially of complex pathways and mechanisms.

As described in our answers to Reviewer 2, we have now added 5 new figures (Figures 4-8) which schematically illustrate the organization of the main ROS responsive signalling pathways that we describe in the text.

  1. Including more specific case studies or examples, particularly in sections discussing the role of ROS in diseases or cellular mechanisms, would provide a more concrete understanding of these processes.

We agree with the reviewer and have now included several additional examples to illustrate key points made in the review.

  1. Ensure that the manuscript includes the latest research findings and technological advancements in the field of ROS research. This might involve discussing new methods for detecting and measuring ROS, recent discoveries in ROS signaling pathways, or novel therapeutic strategies targeting ROS.

We agree with the reviewer that these are certainly relevant issues for the ROS field and so now for example, we have developed a new section (8. ROS in human disease) on lines 754 – 809 which discusses novel therapeutic strategies which target ROS. Furthermore, our descriptions of the main ROS signalling pathways already report many of the latest research findings for these fields. However, we feel that discussing other topics such as methodology and measurement of ROS falls outside of the scope of this current review which is already wide ranging, and so we would prefer to not tackle these issues here.

  1. The evolutionary perspective is intriguing, especially the discussion about fish models. Expanding this section to include a comparative analysis with other model organisms (e.g., mice, fruit flies) could provide a more comprehensive view of ROS mechanisms across different species.

We have now expanded Section 7. (Conservation and Evolution of ROS Signalling) to include some discussion of antioxidant systems in Drosophila in relation to mouse (lines 723 - 737).

  1. A more detailed discussion on current and potential therapeutic strategies targeting ROS, including antioxidants and gene therapy, could be highly valuable. This section could also discuss the challenges and limitations of these therapeutic approaches.

These important point are now covered in a new section of the review (8. ROS in human disease) on lines 754 – 809.

  1. Concluding the manuscript with a section on future research directions would be beneficial. Highlighting unanswered questions and proposing potential research areas could spur further investigation in the field of ROS. This might include emerging fields like the interplay between ROS and epigenetics, or the impact of environmental factors on ROS generation and regulation.

We agree with the reviewer and have now expanded the last part of the final section (9. Conclusions and Perspectives) to more clearly spell out a “wish list” for future research directions in the field – which as the reviewer correctly states, includes the potential involvement of epigenetic regulatory mechanisms (lines 827 – 836).

Reviewer 2 Report

Hong et al. present a review article titled "ROS Signalling and Oxidative Stress: Transcriptional Regulation and Evolution," which delves into the intricate roles of Reactive Oxygen Species (ROS) in cellular physiology and their evolutionary significance. The review commences with a concise overview of oxidative stress, emphasizing the pivotal role of enzymatic and non-enzymatic antioxidant systems in maintaining redox homeostasis. The authors then delve into the dual nature of ROS as both damaging agents and crucial signaling molecules in cellular functions. A particularly intriguing aspect of the review is the discussion on the use of fish as models to study ROS contributions to physiological systems, offering valuable insights into evolutionary perspectives on ROS-mediated transcriptional regulation. Overall, this is an interesting review as it provides a comprehensive overview of ROS signaling and highlights the intricate interplay between evolutionary processes and cellular physiology. However, some points warrant attention before publication:

In the section on Intracellular ROS Balance and Antioxidant Systems, while the authors discuss ROS-mediated oxidation of methionine residues, they should include a brief mention of ROS-mediated oxidation of cysteine residues.

Regarding redox balance, it would be beneficial for the authors to incorporate recent advancements in the field, such as the emergence of reactive sulfur species like hydrogen sulfide, hydropersulfide, and polysulfides as endogenous antioxidants.

In the section on ROS Serve as Signalling Molecules, it would enhance the clarity of the review if the authors include information on the concentrations at which ROS act as signaling molecules and become toxic.

Section 3 could benefit from the inclusion of schematics summarizing the different pathways regulated by ROS, aiding in the visualization of the intricate transcriptional control mechanisms and signaling systems discussed.

In the introduction, it would be advantageous for the authors to briefly elucidate what sets this review apart from existing literature on ROS-Mediated Cellular Signaling, thereby highlighting its unique contributions to the field.

In the section on Intracellular ROS Balance and Antioxidant Systems, while the authors discuss ROS-mediated oxidation of methionine residues, they should include a brief mention of ROS-mediated oxidation of cysteine residues.

Regarding redox balance, it would be beneficial for the authors to incorporate recent advancements in the field, such as the emergence of reactive sulfur species like hydrogen sulfide, hydropersulfide, and polysulfides as endogenous antioxidants.

In the section on ROS Serve as Signalling Molecules, it would enhance the clarity of the review if the authors include information on the concentrations at which ROS act as signaling molecules and become toxic.

Section 3 could benefit from the inclusion of schematics summarizing the different pathways regulated by ROS, aiding in the visualization of the intricate transcriptional control mechanisms and signaling systems discussed.

In the introduction, it would be advantageous for the authors to briefly elucidate what sets this review apart from existing literature on ROS-Mediated Cellular Signaling, thereby highlighting its unique contributions to the field.

Author Response

We would like to thank this reviewer for their very positive and constructive comments. We agree entirely with all the points that they have raised and have adapted the text accordingly.

In the section on Intracellular ROS Balance and Antioxidant Systems, while the authors discuss ROS-mediated oxidation of methionine residues, they should include a brief mention of ROS-mediated oxidation of cysteine residues.

We have now added new text (see lines 78 – 89) which includes this important information.

Regarding redox balance, it would be beneficial for the authors to incorporate recent advancements in the field, such as the emergence of reactive sulfur species like hydrogen sulfide, hydropersulfide, and polysulfides as endogenous antioxidants. In the section on ROS Serve as Signalling Molecules,

We have now included new text (see lines 140 – 155) as part of a new section (3.2: Reactive sulfur species as antioxidants) which corrects this important omission.

In the section on ROS Serve as Signalling Molecules, it would enhance the clarity of the review if the authors include information on the concentrations at which ROS act as signaling molecules and become toxic.

We agree it would be be a good idea to include this extra information and so have added this from lines 182 to 184.

Section 3 could benefit from the inclusion of schematics summarizing the different pathways regulated by ROS, aiding in the visualization of the intricate transcriptional control mechanisms and signaling systems discussed.

We completely agree and so have now added 5 new figures (Figures 4-8) which schematically illustrate the organization of the main ROS responsive signalling pathways that we describe in the text.

In the introduction, it would be advantageous for the authors to briefly elucidate what sets this review apart from existing literature on ROS-Mediated Cellular Signaling, thereby highlighting its unique contributions to the field.

We absolutely agree that this would be a significant improvement for the presentation of our review. For this reason, we have restructured the Introduction section to be able to make this statement clearly and we have more clearely stated the focus of our review in lines 31 – 36.

Reviewer 3 Report

Comments and Suggestions for Authors

This is an interesting, well-written, and well-researched review dealing with the diverse effects of reactive oxygen species (ROS) on cell physiology. Over the past 10-20 years, this subject has been covered in many different reviews of original studies, but more often than not, the emphasis has been on the pro-oxidant detrimental effects of ROS species with relatively little attention given to many of their antioxidant-like signaling effects. This dichotomy is dealt with very well in this manuscript, which is one of its major strengths. Of special interest is the section which deals with the plasticity of ROS signaling in lower animals that have evolved and exist in adverse environments, e.g. fish or rodents living under very cold and/or limited light conditions. This aspect is usually not covered in reviews of this type. Although there is much to like in this offering, some improvements could be made, as suggested in the following comments:

1. It is rather surprising that in the entire discussion of "pro- versus anti-oxidant" effects of ROS nitric oxide (NO) is not mentioned, particularly NO generated naturally by inducible nitric oxide synthase (iNOS). ROS and NO analogues, aka reactive nitrogen oxide species (RNOS) often co-exist in biological systems under oxidative stress conditions. For example, oxidative stress associated with many chemo-, radio-, or photodynamic therapies often causes ROS-induced chain peroxidation of membrane lipids which can be inhibited by iNOS-derived NO at relatively low levels. Inhibition results from NO's ability to intercept and inactivate chain-carrying lipid peroxyl/oxyl radicals. On the other hand, NO (unlike superoxide and hydrogen peroxide) has no enzymatic scavengers and can give rise to byproducts like peroxynitrite, which can induce lipid peroxidation and other damage, depending on where/when such species arise. Thus, NO or RNOS, like ROS, can have diverse anti- vs. pro-oxidant effects, and discussing some of this (with relevant references) would benefit this review. 

2. In Sect. 2, ascorbate's antioxidant properties, i.e. ability to inactivate ROS like superoxide and HO radicals is discussed. However, nothing is said about ascorbate's possible pro-oxidant effects under certain conditions, e.g. when it is near sites of ferric or cupric ion localization. Ascorbate-mediated one-electron reduction of these irons could induce Fenton-like reactions leading to damaging ROS formation. This might occur under pathological conditions associated with iron or copper build-up. Although they may be rare, these possibilities should at least be mentioned in Sect. 2. 

3. Wording suggestions: Although most of the manuscript is carefully worde, there is a "singular" vs. "plural" mix-up in some places, e.g. in sentence on line 187, "ROS" is used in both the singular and plural sense. It is usually meant to be a plural word unless specified otherwise.       

Comments on the Quality of English Language

Use of scientific English is excellent, except for a few minor errors, which are pointed out.

Author Response

We would like to thank this reviewer for their very positive and constructive comments. We agree entirely with all the points that they have raised and have adapted the text accordingly.

  1. It is rather surprising that in the entire discussion of "pro- versus anti-oxidant" effects of ROS nitric oxide (NO) is not mentioned, particularly NO generated naturally by inducible nitric oxide synthase (iNOS). ROS and NO analogues, aka reactive nitrogen oxide species (RNOS) often co-exist in biological systems under oxidative stress conditions………………….

We agree and have now remedied this by including new text that overviews this point (see lines 156 to 172 in the new section 3.3: Role of nitric oxide in oxidative stress).

  1. In Sect. 2, ascorbate's antioxidant properties, i.e. ability to inactivate ROS like superoxide and HO radicals is discussed. However, nothing is said about ascorbate's possible pro-oxidant effects under certain conditions,…………………..

We agree and have now expanded our discussion of the role of ascorbate by including new text that covers this point (see lines 125 to 133).

  1. Wording suggestions: Although most of the manuscript is carefully worde, there is a "singular" vs. "plural" mix-up in some places, e.g. in sentence on line 187, "ROS" is used in both the singular and plural sense. It is usually meant to be a plural word unless specified otherwise.       .

We have now proof-read our text and made corrections and adjustments where necessary. In particular we have corrected sentences where ROS was mistakenly considered a “singular” subject.

Round 2

Reviewer 1 Report

I have already no comments.

Well written

I have already no comments.

Well written

Reviewer 3 Report

The authors agreed that iNOS-derived NO should be included in their review of ROS and RNOS-induced oxidative stress and its signaling effects. Accordingly they have added a section on this in their revised manuscript, which is now much improved and should be ready for publication. 

The manuscript is very well written and requires no significant corrections in wording or interpretation.